# Objective Voice Analysis in Partial Deafness: Comparison of Multi-Dimensional Voice Program (MDVP) and VOXplot Results

**DOI:** 10.3390/jcm13247631

**Published:** 2024-12-14

**Authors:** Karol Myszel

**Affiliations:** 1Center of Hearing and Speech, 7 Mokra Street, 05-830 Kajetany, Poland; karol@myszel.pl; 2Faculty of Health Sciences, University of Applied Sciences, 4 Popieluszko Street, 62-510 Konin, Poland

**Keywords:** MDVP, VOXplot, partial deafness, AVQI, ABI

## Abstract

Acoustic analysis of voice enables objective assessment of voice to diagnose changes in voice characteristics, and track the progress of therapy. In contrast to subjective assessment, objective measurements provide mathematical results referring to specific parameters and can be analyzed statistically. Changes in the voice of patients with partial deafness (PD) were not widely described in the literature, and recent studies referred to the voice parameters measured in this group of patients only using the multi-dimensional voice program (MDVP) by Kay Pentax. This paper describes the results of acoustic analysis of voice in patients with PD using VOXplot, and compares the results with those achieved with MDVP. **Background/Objectives**: The purpose of this study was a VOXplot objective analysis of voice in individuals with PD and to assess consistency with results obtained using MDVP and with perceptual assessment. **Methods**: Voice samples from 22 post-lingual PD individuals were recorded. They included continuous speech (cs) and sustained vowels (sv). The control group consisted of 22 healthy individuals with no history of voice or hearing dysfunction. The samples were analyzed with MDVP followed by VOXplot version 2.0.0 Beta. Statistical analysis was performed using a *t*-test paired with two samples for means. All individuals were also subjected to a perceptual voice assessment using the GRBAS by Hirano. **Results:** Differences were observed in 13 VOXplot parameters measured in voice samples of adults with PD compared with those in the control group. Both multiparametric indices, AVQI and ABI, showed a statistical increase. When it comes to MDVP parameters correlating with breathiness, all of them (shim dB, APQ, NHR, SPI, and NSH) increased in patients with partial deafness, reflecting a breathy voice. Only one increase in the SPI was not statistically significant. Seven MDVP parameters correlating with hoarseness were elevated, and five (Jitt%, vF0, Shim dB, APQ, and NHR) showed a statistically significant increase. Correlations were found of VOXplot and MDVP parameters with perceptual voice assessment. **Conclusions**: Both programs for objective assessment showed voice abnormalities in patients with PD compared with the control groups. There was a poor to moderate level of consistency in the results achieved using both systems. Correlations were also found with GRBAS assessment results.

## 1. Introduction

Partial deafness is described as a condition with normal hearing thresholds at low frequencies (up to 1 kHz) and deep hypoacusis (almost deafness) at higher frequencies. In pure tone audiometry, it is represented as a curve rapidly decreasing at frequencies higher than 1 kHz. Voice dysfunctions associated with partial deafness were not described in the literature until the first such report was published in 2022 based on the research conducted at the Institute of Physiology and Pathology of Hearing in Warsaw. Research is continuing to broaden the scope of our knowledge on the characteristics of voice abnormalities in this type of hearing impairment.

The studies conducted so far were performed using the MDVP system, as this is commonly used in clinical practice. Therefore, comparative analysis of voices in PD patients examined using other systems became an interesting issue. The question of whether findings from other systems were consistent with those of MDVP and reconfirmed voice abnormalities in PD patients tested, became the main rationale for the study described in this paper. In the study, MDVP results were compared with those achieved with the VOXplot system.

VOXplot software 2.0.0 Beta (Lingphon, Straubenhardt, Germany) is used for objective voice assessment in clinical conditions. It focuses on 13 single voice parameters to calculate two main multiparametric indices of hoarseness (AVQI: acoustic voice quality index) and breathiness (ABI: acoustic breathiness index). Validation studies confirmed a strong association between the four main parameters in detecting and describing hoarseness and breathiness. Hoarseness includes the following: (a) harmonic-to-noise ratio (HNR), (b) pitch perturbation quotient for five periods (PPQ5), and breathiness: (a) smoothed central peak prominence (CPPS), (b) glottal-to-noise excitation ratio (GNE) [1].

Based on the specific parameters measured by the VOXplot, two final multiparametric voice quality indices are calculated for voice samples: AVQI, Acoustic Voice Quality Index, which reflects the overall perception of hoarseness and ABI, and the Acoustic Breathiness Index, which reflects the degree of breathiness. AVQI, Acoustic Voice Quality Index calculation is based on HNR, Shimmer%, Shimmer dB, Slope, Tilt and CPPS. Higher values reflect a higher degree of hoarseness. In validated versions of the VOXplot, the AVQI usually achieves values ranging from 0 to 10. As research shows that sex has no influence on AVQI, age has a minimal (almost negligible) influence, and apart from hoarseness, the index may also show voice anomalies related more strongly to breathiness than roughness.

ABI, Acoustic Breathiness Index calculation is derived from CPPS, Jitter%, GNE, HF Noise, HNR-D, H1H2, Shimmer%, Shimmer dB and PSD. Previous studies have shown that ABI has high sensitivity and utility in measuring the degree of breathiness with good diagnostic accuracy. The ABI is not sensitive to changes in age, sex, or roughness. Therefore, its high sensitivity and exclusivity in measuring breathiness makes it a highly valuable tool in clinical practice [2].

Hearing impairments have been widely described in the literature as affecting voice quality. Disturbed auditory control of the voice leads to dysfunction of the vocal folds (establishing abnormal oscillation patterns), respiratory tract, and muscles of the larynx, which results in abnormalities in pitch, volume, and resonance control. The more vocal folds are affected, the greater the distortion of the sound energy. Abnormalities can be objectively measured and detected through perceptual assessments.

The use of VOXplot parameters was accepted by clinical and scientific environments, as they were calculated based on a linear regression model that combined the most relevant parameters. The benefit of using AVQI and ABI indices is also related to the fact that the calculation algorithms are derived from Praat freeware and are based on both continuous speech (cs) and sustained vowels (sv). Descriptions of the VOXplot parameters are listed in Table 1.

The MDVP, a system commonly used in clinical practice, measures 33 parameters of voice samples, recorded as continuous speech and sustained vowels. Parameters are aggregated into seven groups related to (a) frequency, (b) amplitude, (c) noise, (d) tremor (modulation), (e) voice breaks, (f) subharmonics, and (g) irregularities. In MDVP, results are presented in a diagram reflecting the values graphically, as well as in the form of a spectrogram [3,4]. A description of the main MDVP parameters is presented in Table 2.

## 2. Materials and Methods

Voice samples of 22 post-lingual partially deaf individuals (nine females, average age 48.5 years and 13 males, average age 47.7 years) were recorded in sound-isolated anechoic conditions after careful exclusion of preexisting conditions in voice (organic dysphonia, earlier professional voice abuse, respiratory tract diseases, allergies, and neurodegenerative and mental diseases). The samples were recorded as continuous speech (cs) and sustained vowels (sv). The control group consisted of 22 healthy individuals (10 females, average age 54 years and 12 males, average age 40 years) with no history of voice or hearing dysfunction. The samples were analyzed with a multi-dimensional voice program (MDVP) by Kay-Pentax, followed by VOXplot version 2.0.0 Beta. Statistical analysis was performed using a *t*-test paired with two samples for means. All individuals were also subjected to perceptual voice assessment with GRBAS by Hirano, which was performed by two experienced phoniatricians (inter-rater compatibility 92%). The objective measures were then checked for correlations with the subjective features by calculating the Pearson correlation coefficient. This study was approved by the Bioethics Committeee.

## 3. Results

The results of the study showed that major differences can be observed in 13 VOXplot parameters measured in voice samples of adults with PD compared with those in the control group. Decreases were reported in Slope, Tilt, HNR-D, HNR, CPPS, and HF Noise, and all reported changes, excluding HF Noise, were statistically significant.

Other parameters—Shimmer %, Shimmer dB, Jitter local, Jitter ppq5, GNE, and H1H2—showed an increase, and all of the reported changes were statistically significant. Both multiparametric indices showed a statistical increase: AVQI achieved a value of 4.96 (SD = 0.8) vs. 0.35 in the control group, and ABI achieved a value of 6.24 (SD = 0.89) vs. 1.23 in the control group. The reported changes reflect the voices of individuals with partial deafness, such as hoarse and breathy. The average values of the parameters in the patients with partial deafness and those in the control group are presented in Table 3.

Next, a comparison was conducted to check whether the results achieved in the VOXplot analysis were consistent with the MDVP and perceptual analysis. As the literature shows, among all MDVP parameters, Jitt%, vF0, Shim dB, APQ, NHR, SPI, and VTI are strongly associated with the perception of hoarseness, whereas Shim dB, APQ, NHR, SPI, and NSH are associated with the perception of breathy voice and present statistical correlations with breathy voice. To check whether consistency existed between observations performed with VOXplot and MDVP, groups of MDVP parameters correlating with hoarseness and breathiness were compared with VOXplot parameters that most influenced the index of hoarseness (AVQI) and breathiness (ABI), and finally with the final values of the two multiparametric indices themselves. Interestingly, the MDVP analysis showed that all seven parameters correlated with hoarseness were elevated, and five (Jitt%, vF0, Shim dB, APQ, and NHR) showed a statistically significant increase. VOXplot parameters associated with hoarseness in the same group of patients with partial deafness also changed, and all of them presented a statistically significant change indicating hoarseness: Shim % and Shim dB increased, while HNR, Slope, and CPPS decreased. Consistent with the VOXplot methodology, the directions of the aforementioned changes reflected a hoarse voice in patients with PD. The AVQI was also significantly elevated. Therefore, the VOXplot results appeared to be consistent with the MDVP parameters related to hoarseness. Both tools of objective analysis detected statistically significant hoarseness in the voices of partially deaf individuals.

When it comes to MDVP parameters correlating with breathiness, all of them (shim dB, APQ, NHR, SPI, and NSH) increased in patients with partial deafness, reflecting a breathy voice. Only one increase in the SPI was not statistically significant. VOXplot parameters associated with breathy voice also changed, and all but PSD increased significantly. Consistent with the VOXplot methodology, the directions of the above-mentioned changes reflected a breathy voice in PD patients. The results showed that the ABI was significantly elevated. Therefore, the VOXplot results appear to be similar to results obtained with MDVP parameters related to breathiness. Both tools of objective analysis detected a breathy voice in partially deaf individuals. Table 4 presents the MDVP parameters correlating with hoarseness and breathiness compared to the respective VOXplot results.

As the next step, intraclass correlation coefficients (ICCs) were calculated to assess statistical congruence between MDVP and VOXplot parameters that most closely correlate with hoarseness, and subsequently, those that most closely correlate with breathiness. ICC values below 0.5 usually present poor congruence: 0.5 to 0.75 moderate, 0.75 to 0.9 good, and 0.9 to 1.0 excellent congruence. Results obtained in the study are presented in Table 5.

As the data in Table 5 show, moderate reliability between MDVP and VOXplot was observed for hoarseness parameters, and poor reliability was observed for breathiness parameters.

Perceptual assessment of voice in the study group was performed using the Hirano GRBAS scale, which showed hoarseness in 21 of 22 individuals and breathiness in 19 of 22. Hoarseness G1 was present in 19 patients and G2 in two patients. G0 was rated in only one individual and G3 in none. The breathiness of B0 was rated in 3, B1 in 17, and B2 in 2. None of the patients were rated B3. Subjectively, the voices of adults with partial deafness were assessed as being slightly or moderately hoarse and slightly breathy. Correlations of feature G were noted with MDVP parameters (vF0, Shim dB, APQ, SPI) and VOXplot parameters (Shim%, Shim dB, HNR, Slope, Tilt, CPPS, GNE, and H1H2). Correlations of feature B were noted with MDVP parameters (vF0, Shim dB, APQ, NHR, SPI, and NSH) and VOXplot parameters (Shim%, Shim dB, Slope, Tilt, GNE, and H1H2). All voice evaluation methods used in the study of adults with partial deafness revealed hoarse and breathy voice. Table 6 presents the values of the correlation coefficients (Pearson’s R) between the respective objective parameters and G, R, B, A, and S in the perceptual assessment.

## 4. Discussion

Objective voice analysis is a valuable tool for tracking and describing voice changes mathematically. In clinical practice, many different tools are used for objective assessment. The MDVP and Praat systems seem to be the most commonly used; however, VOXplot based on Praat has been receiving more attention recently because of its updated utility, confirmed correlations with coefficients for hoarseness and breathiness, and for enabling measurements using multiparametric models [5].

Research aimed to compare various systems for acoustic voice analysis was conducted by many authors. Research performed in euphonic voices showed inter-gender differences in MDVP and Praat when it comes to absolute jitter, while no differences were present related to shimmer in decibels [6]. Other authors reported similar values of mean fundamental frequency in both systems, with lower values of jitter, shimmer, NHR, and DUV presented in Praat. [7,8]. Shimmer presented a high degree of correlation with HNR in both systems and appeared to be a sensitive parameter in detecting dysphonic voices [9]. Other research showed that higher values of perturbation measures present higher values in MDVP than in Praat, and therefore there cannot be a directly comparison between the systems [10].

Various causes of voice disorders, resulting in dysfunction of the vocal tract or respiratory system, lead to abnormal mass, mobility, and oscillation patterns of the vocal folds. Disturbed vocal function and abnormal air passage can be detected using objective measurements [11]. Therefore, objective voice analysis is commonly used to diagnose different types of voice abnormalities, both organic and functional. For many years, objective voice measures have been recommended for clinical use as a valuable component of voice diagnostics [12,13,14,15,16]. Functional voice disorders associated with hearing impairment have been the subject of many researchers [17,18,19,20,21,22,23,24,25,26,27]. Research performed at the Institute of Physiology and Pathology of Hearing in Warsaw on individuals with partial deafness showed that this group of patients presented with voice abnormalities as a result of inappropriate auditory control of voice production. Voice characteristics discrepancies in people with abnormal hearing are detected in objective measurements using the MDVP tool, particularly those related to frequency, amplitude, noise, and tremor. Perceptually, individuals with partial deafness develop a voice with a small degree of hoarseness, which is slightly harsh, breathy, slightly asthenic, and tense. Analyses have revealed correlations between objective and subjective voice evaluations [28,29,30]. Other research revealed that partial deafness dysphonia is not gender-related; therefore, the study described in this paper did not include a factor of gender.

While MDVP measures specific parameters described individually, VOXplot measures several parameters that come into the final measurement of two multiparametric indices: the acoustic voice quality index (AVQI), which generally describes the level of hoarseness (reflecting disturbed vocal fold oscillation), and the acoustic breathiness index (ABI), which describes voice dysfunction presented with breathiness (reflecting incomplete glottal closure) [12,31,32,33,34].

This study aimed to compare the results of voice analysis conducted using MDVP in patients with partial deafness with those from VOXplot. The acoustic parameters measured by the MDVP are related to frequency, amplitude, noise, tremor, voice breaks, presence of subharmonics, and voice irregularities. In many studies, these parameters have been correlated with subjective voice assessment. Research and observations were conducted to identify which of the objective measures correlated the most with the perceptions of hoarseness and breathiness. A literature review showed that some parameters of frequency (Jita, Jitt, RAP, PPQ, vF0), amplitude (shimmer, APQ), and noise (HNR) are mostly related to hoarseness, whereas the tremor parameter (FTRI) and voice irregularities (DUV) are strongly correlated with harsh voice. Breathy voice in perceptual analysis is strongly associated with amplitude (shimmer, APQ), noise (NHR, SPI), and subharmonics (NSH) [35,36,37,38].

To determine whether objective measurements of voice acoustics performed by the VOXplot corresponded with those performed by the MDVP, parameters related to hoarseness and breathiness were grouped and checked for the levels and directions of changes presented in partially deaf people. The comparison shows that both the MDVP and VOXplot provide quite similar results to the voice acoustics analysis of individuals with partial deafness.

## 5. Conclusions

Both MDVP and VOXplot objective assessments of voice in patients with partial deafness showed abnormalities compared with the control groups.

MDVP measurements revealed statistically significant changes in frequency, amplitude, noise, and tremor parameters. Most of these parameters correlating with the perception of hoarseness and breathiness achieved statistical changes and correlated with the GRBAS findings. VOXplot analysis showed that the increases in the acoustic voice quality index (AVQI) and acoustic breathiness index (ABI) were statistically significant. Correlations were also found with GRBAS assessment results.

Both systems appeared to be sensitive in detecting voice abnormalities in patients with partial deafness. A moderate level of consistency between the systems was found for hoarseness, while a poor consistency level was found for breathiness. Therefore, in clinical practice, measures achieved from both systems should not be directly compared without deeper analysis.

In addition, VOXplot analysis became another tool which reconfirmed the existence of voice dysfunctions in individuals with partial deafness, which is an important reinforcement of the limited number of such observations made so far.

The study described in this paper has some limitations, including the number of participants and the lack of a validated Polish version of VOXplot application. However, it may serve as a preliminary study of the topic and as a foundation for further investigations.

## Figures and Tables

**Table 1 jcm-13-07631-t001:** Descriptions of the parameters measured by VOXplot.

Acoustic Measure Abbreviation	Definition
HNR (dB) (harmonic-to-noise ratio)	Describes the base 10 algorithm of the ratio between the periodic energy and noise energy multiplied by 10 HNR
PPQ5 (%) (jitter of the five-point period perturbation quotient)	Describes the average absolute difference between a period and the average of it and its four closest neighbors divided by the average
CPPS (dB) (smoothed cepstral peak prominence)	Describes the distance between the first harmonic peak and the point with equal quefrency on the regression line through the smoothed cepstrum
GNE (glottal-to-noise excitation ratio)	Describes the glottal-to-noise excitation ratio with a maximum frequency of 4500 Hz
H1H2 (dB) (difference between the first and second harmonics in the spectrum)	Describes the difference between H1 and H2 to localize the first peak and determine F0
HF noise (dB) (high frequency noise)	Describes the relative level of high-frequency noise between the energy from 0 to 6 kHz and energy from 6 to 10 kHz
HNR-D (dB) (harmonic-to-noise ratio from Dejonckere and Lebacq)	Describes the harmonic emergence of the spectral display comprised within the frequency bandwidth between 500 Hz and 1500 Hz
Slope (dB) (general slope of the spectrum)	Describes the difference between the energy within 0–1000 Hz and the energy within 1000–10,000 Hz of the long-term average spectrum
Tilt (dB) (tilt of the regression line through the spectrum)	Describes the difference between the energy within 0–1000 Hz and the energy within 1000–10,000 Hz of the trendline through the long-term average spectrum
PSD (ms) (period standard deviation)	Describes the variation in the standard deviation of periods in which the length of the sample is important for a valid computation of the standard deviation
Jitter local (%)	Describes the average difference between successive periods divided by the average period
Shimmer (%)	Describes the absolute mean difference between the amplitudes of successive periods divided by the average amplitude
Shimmer local (dB)	Describes the base 10 logarithm of the difference between the amplitude of successive periods multiplied by 20

**Table 2 jcm-13-07631-t002:** Descriptions of the parameters measured by MDVP.

Acoustic Measure Abbreviation	Definition
Jita (µs) (absolute jitter)	Describes the absolute change of F0 period
Jitt (%) (jitter percent)	Describes the relative variability of F0
RAP (%) (relative average perturbation)	Describes the relative average perturbation (relative change of F0 with a smoothing factor of 3 periods)
PPQ (%) (pitch period perturbation quotient)	Describes the relative change of F0 with a smoothing factor of 5 periods
sPPQ (%) (smoothed pitch period perturbation quotient)	Describes the relative short and long term changes of F0 with a smoothing factor of 1–199 periods
ShdB (dB) (shimmer in dB)	Describes the relative change of amplitude from period to period (in decibels)
Shim% (shimmer percent)	Describes the relative change of amplitude from period to period (in percent)
APQ (%) (amplitude perturbation quotient)	Describes short term changes of amplitude from cycle to cycle with a smoothing factor of 11 periods
sAPQ (%) (smoothed amplitude perturbation quotient)	Describes the relative changes of amplitude with a smoothing factor of 1–199 periods
vAm (%) (peak amplitude variation)	Describes the relative standard deviation of amplitude from cycle to cycle
NHR (noise-to-harmonic ratio)	Describes the average ratio of non-harmonic energy of the spectrum in 1500–4500 Hz to its harmonic energy in 70–4500 Hz
VTI (voice turbulence index)	Describes the average ratio of non-harmonic energy of the spectrum in 2800–5800 Hz to its harmonic energy in 70–4500 Hz
SPI (soft phonation index)	Describes the average ratio of harmonic energy of the spectrum in 70–1600 Hz to its harmonic energy in 1600–4500 Hz
FTRI (%) (F0 tremor intensity index)	Describes the ratio of frequency of the most intensive modulating component (tremor) to F0 of the sample
ATRI (%) (amplitude tremor intensity index)	Describes the ratio of average amplitude of modulating components in 30–400 Hz to average maximum amplitude
DVB (%) (degree of voice breaks)	Describes the ratio of the total time of voice breaks to the total length of the voice sample
DSH (%) (degree of subharmonics)	Describes the ratio of the number of subharmonic tones to the total number of F0 periods
DUV (%) (degree of voiceless)	Describes the relative number of non-harmonics (without F0) in a total voice sample

**Table 3 jcm-13-07631-t003:** The average values of VOXplot parameters in partial deafness patients and in control group.

	Average in PD Patients	Standard Deviation	Average in Control Group	Standard Deviation	*p*-Value
Slope (dB)	−14.55	4.50	−12.72	3.28	*p* < 0.05
Tilt (dB)	−10.52	1.30	−7.71	1.36	*p* < 0.05
HNR-D (dB)	18.48	1.82	31.79	3.6	*p* < 0.05
HNR (dB)	13.85	4.03	23.91	2.27	*p* < 0.05
Shimmer (%)	7.95	3.21	1.86	1.25	*p* < 0.05
Shimmer (dB)	0.80	0.25	0.27	0.37	*p* < 0.05
CPPS (dB)	6.26	1.49	19.21	1.47	*p* < 0.05
Jitter local (%)	1.40	0.68	0.21	0.11	*p* < 0.05
Jitter ppq5 (%)	0.65	0.33	0.14	0.07	*p* < 0.05
GNE	0.95	0.03	0.89	0.06	*p* < 0.05
HF Noise	1.26	0.23	1.38	0.34	*p* > 0.05
H1H2	3.15	2.36	1.53	2.46	*p* < 0.05
PSD	0.84	0.67	0.38	0.57	*p* > 0.05
AVQI	4.96	0.80	0.35	0.67	*p* < 0.05
ABI	6.24	0.89	1.23	0.48	*p* < 0.05

**Table 4 jcm-13-07631-t004:** MDVP parameters correlating with hoarseness and breathiness compared with respective VOXplot results.

	MDVP				VOXplot			
		Partial Deafness	Control	*p* Value		Partial Deafness	Control	*p* Value
Hoarseness	Jitt %	1.84	0.40	<0.05	Shim%	7.95	1.86	<0.05
	vF0	8.4	0.74	<0.05	Shim dB	0.8	0.27	<0.05
	Shim dB	0.73	0.27	<0.05	HNR	13.85	23.91	<0.05
	APQ	6.41	1.8	<0.05	Slope	−14.55	−12.72	<0.05
	NHR	0.2	0.12	<0.05	Tilt	−10.52	−7.77	<0.05
	SPI	10.31	8.72	>0.05	CPPS	6.26	19.21	<0.05
	VTI	0.06	0.04	>0.05	AVQI	4.96	0.35	<0.05
Breathiness	Shim dB	0.73	0.27	<0.05	Jitter%	1.4	0.21	<0.05
	APQ	6.41	1.8	<0.05	Shim dB	0.8	0.27	<0.05
	NHR	0.2	0.12	<0.05	GNE	0.95	0.89	<0.05
	SPI	10.31	8.72	>0.05	H1H2	3.15	1.53	<0.05
	NSH	0.55	0	<0.05	PSD	0.84	0.38	>0.05
					ABI	6.24	1.23	<0.05

**Table 5 jcm-13-07631-t005:** Values of intraclass correlation coefficient between MDVP and VOXplot related to hoarseness and breathiness parameters.

MDVP parameters correlating with hoarseness	VOXplot parameters correlating with hoarseness	Intraclass correlation coefficient (ICC)
vF0, Jitt%, Shim dB, APQ, NHR, SPI, VTI	Shim%, Shim dB, HNR, Slope, Tilt, CPPS	0.56
MDVP parameters correlating with breathiness	VOXplot parameters correlating with breathiness	Intraclass correlation coefficient (ICC)
Shim dB, APQ, NHR SPI, NSH	Jitter%, Shim dB, GNE, H1H2, PSD	0.47

**Table 6 jcm-13-07631-t006:** Values of correlation coefficient (R Pearson) between respective objective parameters and perceptual assessment.

	Parameter	G	R	B	A	S
MDVP	Jitt %	-	0.24	-	0.24	-
	vF0	0.68	0.29	0.38	-	0.3
	Shim dB	0.74	-	0.46	-	-
	APQ	0.25	-	0.20	-	-
	NHR	-	-	0.78	0.24	0.4
	SPI	0.24	-	0.28	0.24	-
	VTI	-	0.26	-	0.26	0.29
	NSH	-	0.27	0.88	0.4	-
VOXplot	Shim%	0.49	-	0.46	0.47	0.38
	Shim dB	0.34	0.4	0.23	0.51	-
	HNR	0.46	0.24	-	0.26	-
	Slope	0.30	-	0.32	-	0.25
	Tilt	0.47	-	0.44	-	0.35
	CPPS	0.47	0.24	-	0.39	-
	Jitter%	-	0.43	-	0.42	-
	GNE	0.50	0.26	0.35	0.34	0.3
	H1H2	0.67	-	0.46	-	0.45
	PSD	-	0.39	-	0.34	0.45

## Data Availability

The data presented in this study are available on request from the author.

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
