# Peer review of "Objective Voice Analysis in Partial Deafness: Comparison of Multi-Dimensional Voice Program (MDVP) and VOXplot Results"

_jcm, 2024, doi:10.3390/jcm13247631_

Round 1

Reviewer 1 Report

Comments and Suggestions for Authors

General comments: The authors have made an attempt to study the efficiency of VOXplot software for voice analysis of clinical population, specifically on individuals with partial deafness. The VOXplot is relatively a new Open software for analysis of voice across multiple parameters and provides multiple indices which can be used in the diagnosis of voice disorders. However, the authors need to focus on following aspects:

1.     The manuscript is lacking adequate review in introduction section. The introduction section just limited to the description of various voice analysis parameters which are provided by the softwares and I feel this information is not necessary in a research manuscript as most of researchers and clinicians are aware about these parameters.

2.     If the objective of the study is compare the results of two softwares, the authors should provide more review on these aspects with the differences between multiple softwares available. There are quite a few studies on comparing the efficacy of various softwares in voice analysis. The following studies may help authors:

·        Amir O, Wolf M, Amir N. A clinical comparison between two acoustic analysis softwares: MDVP and Praat. Biomedical Signal Processing and Control. 2009 Jul 1;4(3):202-5.;

·        Lovato A, De Colle W, Giacomelli L, Piacente A, Righetto L, Marioni G, de Filippis C. Multi-Dimensional Voice Program (MDVP) vs Praat for assessing euphonic subjects: a preliminary study on the gender-discriminating power of acoustic analysis software. Journal of Voice. 2016 Nov 1;30(6):765-e1.;

·        Yoo JY, Jeong OR, Jang TY, Ko DH. A correlation study among acoustic parameters of MDVP, Praat, and Dr. Speech. Speech Sciences. 2003;10(3):29-36.;

·        Maryn Y, Corthals P, De Bodt M, Van Cauwenberge P, Deliyski D. Perturbation measures of voice: a comparative study between Multi-Dimensional Voice Program and Praat. Folia Phoniatrica et Logopaedica. 2009 Jul 9;61(4):217-26.

·        Ko HJ, Woo MR, Choi Y. Comparisons of voice quality parameter values measured with MDVP, Praat, and TF32. Phonetics and Speech Sciences. 2020;12(3):73-83.

3.     The introduction section does not contain JUSTIFICATION FOR THE STUDY nor the objectives of the study. The authors may look into this section more carefully rather than just describing the parameters.

4.     The manuscript does not have adequate information about the need for this study on individuals with partial deafness (post-lingual). Are there any supporting studies which reported voice abnormalities in post-lingual individuals with partial deafness? It would be more appropriate if the authors include participants with any specific voice disorders such as organic voice disorders to make the study more effective.  

5.     The study involved a total of 9 females and 13 males in clinical group & a total of 10 females and 12 males in control group. However, the results section shows that the analysis was done without the factor of gender. Can the authors describe why there was no comparison of results with gender as a factor as we all are aware that voice characteristics can vary between genders?

6.     AVQI & ABI is significantly higher in PD group with more hoarse and breathy voices. Is it correlated with GRBAS scale? Do you have any previous studies on voice characteristics of partial deafness with such high AVQI or ABI indices? 

7.     In the literature, we have the two terms of hoarseness and breathiness with a clear demarcation in perceptual analysis of voice. However, the manuscript reported that the majority of the patients have been reported with both hoarseness and breathiness. Can the authors describe about it in more detail? Usually we see hoarseness in patients with hyperfunctional voice disorders with intermittent voicing and breathiness is seen in bilateral vocal fold paralysis with no voicing.  

8.     How the parameters of MDVP was were divided between hoarseness and breathiness? The question here is that some of the features are included in both of them.

9.     The authors may relook into the analysis of GRBAS ratings and more details should be included about these ratings on other parameters such as Roughness, and Strained. The Grade (G) refers to the overall abnormality of voice or the grade of hoarseness and be rated based on the remaining four parameters.

1 - The sample size included in the study may not be sufficient to draw conclusions about the validty of VOXplot software while comparing it with MDVP

Author Response

  1. The manuscript is lacking adequate review in introduction section. The introduction section just limited to the description of various voice analysis parameters which are provided by the softwares and I feel this information is not necessary in a research manuscript as most of researchers and clinicians are aware about these parameters. Answer: the introduction section has been changed. Description of acoustic methods was shortened and limited to the introductory level only, avoiding detailed definitions. Besides, general information on the rationale for the study was added.
  2.  If the objective of the study is compare the results of two softwares, the authors should provide more review on these aspects with the differences between multiple softwares available. There are quite a few studies on comparing the efficacy of various softwares in voice analysis. Answer: as suggested, some comparative information on other authors` findings related to MDVP and Praat were added in the discussion section. Recommended publications were used and added to References.
  3. The introduction section does not contain JUSTIFICATION FOR THE STUDY nor the objectives of the study. The authors may look into this section more carefully rather than just describing the parameters. Answer: a rationale for the study (purpose justification) was added in the Introduction section. Desription of parameters were shortened.
  4. The manuscript does not have adequate information about the need for this study on individuals with partial deafness (post-lingual). Are there any supporting studies which reported voice abnormalities in post-lingual individuals with partial deafness? It would be more appropriate if the authors include participants with any specific voice disorders such as organic voice disorders to make the study more effective. Answer: the need was described (broadening our knowledge on voice disorders in partial deafness). The author is the first researcher worldwide who described voice disorders associated with partial deafness (3 prior publications also mentioned in References and described in Introduction and Discussion sections). This current paper is a development of the research. Partial deafness is a rersearch topic since 2002 (first cochlear implantation in this type of hearing impairment). Since this time it has been a research topic in many aspects. The reason to exclude other voice disorders (organic) was that I wanted to examine how partial deafness itself influences voice quality, therefore other voice dysfunctions were excluded purposely.
  5. The study involved a total of 9 females and 13 males in clinical group & a total of 10 females and 12 males in control group. However, the results section shows that the analysis was done without the factor of gender. Can the authors describe why there was no comparison of results with gender as a factor as we all are aware that voice characteristics can vary between genders? Answer: in other recent study of mine (recently approved for publication in Journal of Voice, but no DOI assigned yet) I proved that partial defaness dysphonia is not gender-related. This was the main reason not to use the gender factor in this current study. Besides, the VOXplot authors proved no significant dependency of AVQI and ABI on gender (described in the Introduction). 
  6.   AVQI & ABI is significantly higher in PD group with more hoarse and breathy voices. Is it correlated with GRBAS scale? Do you have any previous studies on voice characteristics of partial deafness with such high AVQI or ABI indices? Answer: in prevoius studies voice of patients with partial deafness has been analyzed with MDVP only. This current study is the first one using VOXplot in this group of patients, so AVQI and ABI were not measured before. Both MDVP and VOXplot analysis showed voice dysfunctions in PD patients. It is correlated with GRBAS scale both in this and in previous studies.
  7. In the literature, we have the two terms of hoarseness and breathiness with a clear demarcation in perceptual analysis of voice. However, the manuscript reported that the majority of the patients have been reported with both hoarseness and breathiness. Can the authors describe about it in more detail? Usually we see hoarseness in patients with hyperfunctional voice disorders with intermittent voicing and breathiness is seen in bilateral vocal fold paralysis with no voicing. Answer: As my previous studies show, voices of PD patients feature functional dysfunctions presented with hoarseness and breathiness. Videolaryngoscopic findings also show discoordination of vocal fold mobility as well as incomplete vocal closure in majority of patients. Hoarseness, depending on vocal fold mobility, vocal tension and vocal mass, occurs in PD patients quite often as a result of inappropriate mobility and tension (discoordinatin of phonation and respiratory mechanics due to disordered auditory control). Breathiness, resulting from incomplete closure coexists with hoarseness. As written in the paper, PD patients usually develop hoarseness G1 and breathiness B1 (reference: Myszel K, Szkielkowska A. Quality of voice in patients with partial deafness. J Voice, June 3, 2022).
  8. How the parameters of MDVP was were divided between hoarseness and breathiness? The question here is that some of the features are included in both of them. Answer: based on the literature, MDVP parameters were divided into two groups according to their highest correlations with hoarseness and breathiness. Some publications, including mine related to partial deafness, described the correlations. Paremeters were therefore selected according to the "power" of their influence on hoarseness and breathiness. The description is a part of the Results section.
  9. The authors may relook into the analysis of GRBAS ratings and more details should be included about these ratings on other parameters such as Roughness, and Strained. The Grade (G) refers to the overall abnormality of voice or the grade of hoarseness and be rated based on the remaining four parameters. Answer: the remaining features R, A and S were included in the analysis and calculation of correlations was added.
  10. The sample size included in the study may not be sufficient to draw conclusions about the validty of VOXplot software while comparing it with MDVP. Answer: in the final part of the Conclusions a note was added on limitations of the study together with the need of further investigation. The study is a preliminary work and VOXplot needs a Polish language validation on a bigger number of subjects (which is now under process). 

I need to express my gratitude to the Reviewer for very precious, meritorical comments and suggestions. I am positive, that my work benefits a lot from them, becoming more scientific and comprehensive.

Reviewer 2 Report

Comments and Suggestions for Authors

The work is a very worthwhile example to show how little investment of time in computing objective voice quality measures can help gain deeper understandings in many fields or research. 

Minor points that should be addressed: AVQI and ABI are modern measurements on which slowly but consistently more work is published. For the value of these measurements it is of no importance, whether they are measured by their author's software in Praat or recently published free software in VoxPlot. The names AVQI and ABI should therefor appear in the abstract (or at least the keywords) in order to allow search engines to find this paper when scientists or doctors search for them.  

The term "partially deaf" seems important to understand this work an some kind of definition of this term should be given. Who is included in this and who is not?

One of the more important claims of the paper is that "There is a high consistency of results between MDVP and VOXplot". However, I see no formal examination of that claim. Computing correlation coefficients (e.g. ICC or other measures of congruency might substantiate the clain and give data for the design of further studies. Personally, I would even prefer a good visualization/plot so the readers get an impression on how consistent the measures are. 

Apart from that the work is well written and I like tables 1 and 2 a lot.

Author Response

  1. Minor points that should be addressed: AVQI and ABI are modern measurements on which slowly but consistently more work is published. For the value of these measurements it is of no importance, whether they are measured by their author's software in Praat or recently published free software in VoxPlot. The names AVQI and ABI should therefor appear in the abstract (or at least the keywords) in order to allow search engines to find this paper when scientists or doctors search for them. Answer: as suggested, the names AVQI and ABI were put in the Keywords sections. Definitions were only shortly written in the Introduction part.
  2. The term "partially deaf" seems important to understand this work an some kind of definition of this term should be given. Who is included in this and who is not? Answer: the definiton of partial deafness was added in Introduction. Clearly, this was missing. 
  3. One of the more important claims of the paper is that "There is a high consistency of results between MDVP and VOXplot". However, I see no formal examination of that claim. Computing correlation coefficients (e.g. ICC or other measures of congruency might substantiate the clain and give data for the design of further studies. Personally, I would even prefer a good visualization/plot so the readers get an impression on how consistent the measures are. Answer: to make the statements more objective and measurable, intraclass correlation coefficients (ICCs) were calculated and presented in a separate table. Therfore, the levels of consistency (reliability) were made objective to draw more reliable coclusions based on them. This was a crucial comment, that made my work results more comprehensive and objective.

I need to express my special gratitude to the Reviewer for the comments and suggestions that helped to give much more scientific value to my paper.